# Persistent Depressive Disorder-Related Effect of Sleep Disorder on the Highest Risk of Suicide in Taiwan, 2000–2015

**DOI:** 10.3390/ijerph192013169

**Published:** 2022-10-13

**Authors:** Sheng-Huang Hsiao, Chih-Chien Cheng, Iau-Jin Lin, Chia-Peng Yu, Yao-Ching Huang, Shi-Hao Huang, Chien-An Sun, Li-Yun Fann, Miin-Yea Sheu, Wu-Chien Chien

**Affiliations:** 1Department of Neurosurgery, Taipei City Hospital, Ren-Ai Branch, Taipei 10629, Taiwan; 2Department of Psychology, National Chengchi University, Taipei 11605, Taiwan; 3Department of Mechanical Engineering, National Central University, Jhongli 32001, Taiwan; 4Department of Obstetrics/Gynecology, Taipei City Hospital, Taipei 10341, Taiwan; 5School of Medicine, College of Medicine, Fu-Jen Catholic University, New Taipei City 242062, Taiwan; 6Department of Medical Research, Tri-Service General Hospital, Taipei 11490, Taiwan; 7School of Public Health, National Defense Medical Center, Taipei 11490, Taiwan; 8Department of Chemical Engineering and Biotechnology, National Taipei University of Technology (Taipei Tech), Taipei 10608, Taiwan; 9Department of Public Health, College of Medicine, Fu-Jen Catholic University, New Taipei City 242062, Taiwan; 10Big Data Center, College of Medicine, Fu-Jen Catholic University, New Taipei City 242062, Taiwan; 11Department of Nursing, Taipei City Hospital, Taipei 10684, Taiwan; 12Department of Nurse-Midwifery and Women Health, National Taipei University of Nursing and Health Sciences, Taipei 11220, Taiwan; 13Graduate Institute of Life Sciences, National Defense Medical Center, Taipei 11490, Taiwan; 14Taiwanese Injury Prevention and Safety Promotion Association (TIPSPA), Taipei 11490, Taiwan

**Keywords:** persistent depressive disorder (PDD), sleep disorder (SD), suicide, poor prognosis

## Abstract

Objective: to investigate whether persistent depressive disorder (PDD) affects sleep disorders (SDs) and increased suicide risk. Methods: in this study, we used the National Health Insurance Research Database (NHIRD) to select 117,033 SD patients, of whom 137 died by suicide, and 468,132 non-SD patients, of whom 118 died by suicide, and analyzed gender, age, and co-existing diseases. Hazard ratios (HRs) and 95% confidence intervals (CI) were calculated using a multivariate Cox proportional hazards model. Results: the hazard ratio of suicide in SD patients was 1.429 times that of non-SD patients. The hazard ratio of suicide in female patients was 1.297 times higher than in males. Compared with people without PDD, people with PDD had a 7.195 times higher hazard ratio for suicide than those without PDD. PDD patients with SDs had a 2.05 times higher hazard ratio for suicide than those with no SDs. Conclusions: suicide risk was increased in SD patients, and the maximum suicide risk was greater in SD patients with PDD than in non-PDD patients. PDD affected SDs and increased suicide risk. Clinicians should be aware of the possibility that PDD affects patients with SDs and contributes to suicide risk.

## 1. Introduction

Sleep disturbances are becoming more common around the world, whether caused by health problems or by excessive stress [1]. Sleep disorders (SDs) are a group of disorders that affect the ability to sleep on a regular basis, as well as medical conditions that affect the quality and duration of sleep [2]. The sleep deprivation caused by SDs can have a major impact on daytime routine, quality of life, and overall health [3]. Most people experience sleep problems due to stress, busy schedules, and other external influences [4]. SDs can cause people to have difficulty falling asleep, and they may thus feel very tired throughout the day [5]. In some cases, sleep disturbance may be a symptom of another medical or mental health condition [6]. When sleep disturbances are not caused by other medical conditions, treatment typically involves a combination of medication and lifestyle changes [7]. Evidence suggests that the link between sleep deprivation and suicide is relatively direct, whereas the link between excess sleep and suicide appears to be relatively indirect (i.e., excess sleep is driven by underlying chronic health conditions and vice versa) [8].

The World Health Organization (WHO) has estimated that 800,000 people die by suicide worldwide each year [9]. Suicide is caused by a complex combination of social risk factors, demographic characteristics, interdependencies (i.e., co-occurring disorders such as mental illness and substance use), and multiple levels of causality [10]. Some suicides are impulsive actions due to stress (such as financial or academic difficulties), relationship problems (such as a breakup), or harassment, bullying, and so on [11,12,13]. People who have attempted suicide have a relatively high risk of future suicide attempts [11]. However, anxiety, depression, eating and trauma-related disorders, and organic mental disorders also contribute to suicide [14]. Well-adjusted emotional regulation is the foundation of human life and mental health [15]. Negative physiological emotions are offset by positive emotions, and this balance is the backbone of the human physiological/emotional state [16]. However, when negative emotions are not properly balanced, this mechanism can sometimes become dysfunctional, leading to maladaptive behaviors, especially during adolescence [15,16]. An overview of current reports indicates that suicide is a highly complex and multifaceted phenomenon in which a large number of mechanisms may be involved, especially in individuals with major depressive disorder (MDD) [17].

Persistent depressive disorder (PDD) has a strong relationship with suicide; however, it lacks specificity as a predictor, and little is known about the characteristics that increase suicide risk in people with depression [18]. PDD is known to be the most common disorder among those who die by suicide, and previous studies have shown that risk factors for the disorder include a family history of mental disorders, male sex, suicide attempts, more severe depression, hopelessness, and comorbidities [19]. Currently, there is limited information regarding suicide risk in depression, reflecting a surprisingly low amount of research given the high suicide risk associated with the disorder [18,19].

As many as 90% of people with PDD report having SD complaints [20]. Insomnia occurs in approximately two-thirds of patients with major depressive episodes, and approximately 40% complain of difficulty falling asleep, frequent awakenings, or early morning awakening (delayed or end-stage insomnia), all three of which are reported by many patients [21,22]. The bidirectional association between SDs and PDD increases the difficulty of distinguishing causal relationships between them [23]. Nineteen studies met the inclusion criteria [24]. Compared with patients with no SDs, patients with a psychiatric diagnosis and comorbid SD were significantly more likely to report suicidal behavior (OR = 1.99, 95% CI 1.72, 2.30, *p* < 0.001). This association was also seen in several psychiatric disorders, including depression (OR = 3.05, 95% CI 2.07, 4.48, *p* < 0.001), post-traumatic stress disorder (PTSD; OR = 2.56, 95% CI 1.91, 3.43, *p* < 0.001), panic disorder (OR = 3.22, 95% CI 1.09, 9.45, *p* = 0.03), and schizophrenia (OR = 12.66, 95% CI 1.40, 114.44, *p* = 0.02) [24,25]. A better understanding of how PDD affects SD in terms of the risk of suicide can aid public health efforts. Currently, there are limited longitudinal observational studies on the relationship between how PDD affects SDs in terms of the risk of suicide. Therefore, we hypothesized that PDD affects SDs in terms of having the highest risk of suicide. We used the Ministry of Health and Welfare’s National Health Insurance Research Database (NHIRD) to track whether PDD affected SDs in terms of the risk of suicide from 2000 to 2015 in Taiwan through long-term follow-up.

## 2. Method

### 2.1. Data Sources

This study was based on the NHIRD of Taiwan. Currently, the National Health Insurance (NHI) program includes information for most of Taiwan’s population. The NHIRD contains multiple health registries for most Taiwanese populations, including outpatient, inpatient, and emergency department data, as well as physicians’ code diagnoses according to the International Classification of Diseases, Ninth Clinical Revision (ICD-9-CM). The NHIRD has high accuracy and validity in terms of disease diagnosis. Therefore, it provides representative data for medical-related research. In addition, the NHIRD encrypts and converts all identification numbers in all database records to ensure the privacy of everyone who enrolls in the program. Our study was approved by the Tri-Services General Hospital Institutional Review Board (TSGHIRB: No. B-110-55).

### 2.2. Study Design

This study was a population-based cohort tracking study that recruited 1,978,082 outpatients and inpatients from the Taiwan Longitudinal Health Insurance Database (LHID) between 1 January 2000 and 31 December 2015. We included people diagnosed with SDs (783,918 individuals; ICD-9-CM codes 780.5 and 780.50) and who had at least three outpatient visits or one hospitalization. Patients with SDs and injuries diagnosed before 1 January 2001, those who had fewer than three outpatient visits, those injured before tracking, those whose gender was unknown, those under 20 years of age, and those with Match < 4 were excluded. The index date was defined as the date of a newly diagnosed SD. In total, 117,033 participants (137 of whom died by suicide) who met our inclusion criteria were assigned to the study cohort. A control group consisting of 468,132 undiagnosed SD patients (118 of whom died by suicide) was randomly selected and paired with the SD group by age, sex, index date, and comorbidities using a 1:4 ratio. Among control patients, patients with a history of SD and suicide during the study were excluded according to our exclusion criteria. A flowchart of the study design is shown in Figure 1.

### 2.3. Outcomes

In this study, we followed all participants from the censoring day until their suicide or through 31 December 2015. We classified suicide mechanisms according to ICD-9 diagnoses. Comorbidities include diabetes mellitus (DM), hypertension (HTN), PDD, chronic kidney disease (CKD), heart failure (HF), chronic obstructive pulmonary disease (COPD) and related disorders, dyslipidemia, and diffuse connective tissue disease. The baseline characteristics of the two groups were compared. To correct the difference in the degree of urbanization between the violent abuse cases and the areas where the control group lived, the degree of urbanization was defined based on the counties, cities, towns, and urban areas in Taiwan corresponding to the area codes of the insured units in the insurance data file. According to Liu Jieyu’s research in 2006, Taiwan’s 359 townships (excluding outlying islands) were classified according to population density (person/km^2^), proportion of the population with a college education or above (%), proportion of the population over 65 years old (%), agricultural proportion of the population (%), the number of Western doctors per 100,000 people, and five other variables. Seven clusters were identified using the cluster analysis method and these were named as follows: cluster 1: highly urbanized towns; cluster 2: moderately urbanized towns; cluster three: emerging towns; cluster four: general towns and urban areas; cluster five: aging towns; cluster six: agricultural towns; and cluster seven: remote towns [26]. The categorization of low income was based on the standards published by the Central Department of Budget, Accounting, and Statistics and was defined by the central and municipality authorities as 60% of the median personal expenditure amount in the household’s local area in the past year [27].

### 2.4. Statistical Analysis

In this study, we compared demographic characteristics and common comorbidities between SD and non-SD patients using the chi-squared test. Average patient age in both cohorts was calculated using Student’s *t*-test. The injury incidence (per 10^5^ person-years) was calculated based on sex, age, and comorbidities for each cohort. Adjustments were made for age, sex, and concomitant comorbidities for inclusion in the multivariate model. Hazard ratios (HRs) and 95% confidence intervals (CI) were calculated using a multivariate Cox proportional hazards model. All statistical analyses were performed using SPSS 22.0 software, and a *p*-value < 0.05 was considered statistically significant.

## 3. Results

### 3.1. Baseline Characteristics of the Patients in the Study

The baseline demographic characteristics and common comorbidity data of the patients are shown in Table 1. The study included 117,033 patients in the SD cohort and 468,132 patients in the control group between 2000 and 2015. The average age of the SD cohort was 53.34 ± 15.7 years, and the proportion of female patients was 51.8%. Among the study population, 55.2% of patients were 40–64 years old, 22.7 were 65 years and older, and 22.0% were 20–39 years old. The distribution of low income, location, hospital level, and comorbidities (DM, HTN, depressive neurosis, CKD, HF, COPD and related disorders, dyslipidemia, and diffuse connective tissue disease) between study and control groups was significant (*p*-value < 0.001).

### 3.2. Factors of Suicide at the End of Follow-Up Using Cox Regression

Table 2 shows the results of the Cox regression analysis of the hazard ratio for the SD and control cohorts. After adjusting for age, sex, comorbidities, geographical area of residence, urbanization level of residence, and low income, the suicide hazard ratio for those with SDs was 1.429 times higher than for those with no SDs. Females were 1.297 times more likely to die by suicide than males. Age was a protective factor for suicide (HR = 0.946, 95% CI: 0.935–0.957, *p* < 0.001). People with PDD had 7.195 times the hazard ratio for suicide compared with those without PDD. The hazard ratio for suicide in winter was 0.063 times higher than in spring. The hazard ratio for suicide in a medical center, regional hospital, or local hospital was 15.208 times, 21.651 times, and 23.242 times higher than in clinics, respectively (*p*-value < 0.001).

### 3.3. Comparison of the Risk of Suicide between the Sleep Disorder and Control Cohorts

We stratified and analyzed sex, age, and concomitant comorbidities for the risk of injury using the Cox regression model, as presented in Table 3. After adjusting for age, sex, and other concomitant comorbidities, SD patients had a 42.9% higher risk of suicide (adjusted HR = 1.429 (95% CI, 1.073–1.905); *p*-value = 0.015). Males with SDs had 2.104 times the risk of suicide compared with those with no SDs (adjusted HR = 2.104 (95% CI, 1.330–3.329); *p*-value = 0.002). People with SDs aged 40–64 had a 1.924 times higher risk of suicide than those with no SDs (adjusted HR = 1.924 (95% CI, 1.293–2.862); *p*-value = 0.001). People with SDs in non-low-income households had 1.484 times the risk of suicide compared with those with no SDs (adjusted HR = 1.484 (95% CI, 1.112–1.979); *p*-value = 0.007). SD patients with DM had 1.868 times the risk of suicide compared with those with no SDs (adjusted HR = 1.868 (95% CI, 1.053–3.312); *p*-value = 0.033). People with SDs and no HTN had 1.655 times the risk of suicide compared with those with no SDs (adjusted HR = 1.655 (95% CI, 1.140–2.401); *p*-value = 0.008). SD patients with PDD had 2.050 times the risk of suicide compared with those with no SDs (adjusted HR = 2.050 (95% CI, 1.350–3.112); *p*-value = 0.001). SD patients with no CKD had 1.483 times the risk of suicide compared with those with no SDs (adjusted HR = 1.483 (95% CI, 1.104–1.993); *p*-value = 0.009). SD patients with no HF had 1.447 times the risk of suicide compared with those with no SDs (adjusted HR = 1.447 (95% CI, 1.068–1.959); *p*-value = 0.017). SD patients with no COPD had 1.769 times the risk of suicide compared with those with no SDs (adjusted HR = 1.769 (95% CI, 1.207–2.592); *p*-value = 0.004). SD patients with no disorders of lipoid metabolism had 1.715 times the risk of suicide compared with those with no SDs (adjusted HR = 1.715 (95% CI, 1.207–2.437); *p*-value = 0.003). SD patients with no diffuse diseases of the connective tissue had 1.526 times the risk of suicide compared with those with no SDs (adjusted HR = 1.526 (95% CI, 1.137–2.046); *p*-value = 0.005). The suicide risk in autumn was 7.259 times higher for those with SDs than for those with no SDs (adjusted HR = 7.259 (95% CI, 1.824–28.884); *p*-value = 0.005). The risk of suicide in patients with SDs residing in a location at urbanization level 3 was 2.071 times that of those with no SDs (adjusted HR = 2.071 (95% CI, 1.219–3.518); *p*-value = 0.007). Finally, the suicide risk was 3.319 times higher for those with SDs than those with no SDs in clinics (adjusted HR = 3.319 (95% CI, 1.160–9.498); *p*-value = 0.025).

## 4. Discussion

The results of this study show that the suicide risk in those with SDs was 1.429 times higher than in those with no SDs. Males were 1.297 times more likely to die by suicide than females. Age was a protective factor for suicide. People with PDD exhibited a 7.195 times higher risk of suicide compared with those without PDD. SD patients with PDD exhibited a 1.174 times higher risk of suicide compared with those with no SDs. The link between SDs and depression is strong [28]. Most people who have experienced depression know that it is often accompanied by SD problems [29]. People with depression may find it difficult to fall asleep and stay asleep at night [30]. They may also have excessive daytime sleepiness or may even sleep too much [31]. At the same time, SDs can exacerbate depression, leading to a negative cycle between depression and sleep that can be difficult to break [28]. The phenomenology of sleep and PDD has demonstrated that disturbances in sleep continuity are ubiquitously accompanied by affective disturbances [32].

WHO showed that depression can lead to suicide, and more than 700,000 people die by suicide every year [33]. Suicide is the fourth-leading cause of death among 15–29-year-olds [9]. Depression is associated with suicidal tendencies, and in most countries, suicide is one of the most common causes of death in people with depression [34]. More specifically, estimates suggest that approximately 60% of suicide victims suffer from MDD and other mood disorders [35]. Male norms can be detrimental to the mental health of young men, putting them at greater risk for suicidal ideation [36]. Mood disturbances (e.g., recurrent depressive disorder or depressive episodes) may be important factors leading to negative assessments of one’s ability to cope with difficult situations and a greater tendency to view stressful events as overwhelming, ultimately leading to suicide [37]. People with depression use ineffective and avoidant strategies to cope with stress more often than healthy people [38].

Previous studies showed that the potential link between SDs and suicide has been the subject of several reviews, and one meta-analysis was conducted to estimate the overall association between SDs and suicidal behavior and to identify more specific relationships among patients with depression [39]. The relationship between PDD and SDs leading to increased suicide risk has not been established. Several studies have been conducted that may help to explain the underlying mechanism. Some studies have suggested that the activity of serotonin (5-HT) is the main cause of the association between SDs and suicide [40,41]. 5-HT is an important neurotransmitter that promotes wakefulness and the onset of sleep through the persistent inhibition of slow-wave sleep (SWS) and rapid eye movement sleep (REM), and its dysfunction may lead to SDs [42]. Depressed patients were observed to have decreased SWS and decreased concentrations of 5-hydroxyindoleacetic acid (5-HIAA, the major metabolite of 5-HT) in their cerebrospinal fluid. Benson et al. reported that SWS in individuals with depression was closely related to serotonin function [43]. These results suggest that serotonergic dysfunction may be a risk factor for SDs. Furthermore, retransferring a specific 5-HT2 antagonist has been shown to increase SWS in depression [44], confirming the hypothesis that 5-HT plays an important role in the regulation of sleep processes. Furthermore, a reduction in 5-HIAA has been shown to be associated with depression, modulating impulse control and acting as a risk marker for suicidal behavior [45]. Therefore, abnormal serotonin function is considered an important physiological factor in the association between SDs and suicidal behavior in depressed patients [46].

However, McCall et al. suggested that specific SDs do not necessarily lead to suicide through serotonin [47]. They propose that SDs disrupt the functioning of serotonin [47]. SDs appear to be indirectly associated with suicide risk because of imbalanced beliefs and attitudes toward sleep and nightmares [46]. Bernert et al. demonstrated that less non-REM stage 4 sleep and a higher frequency of nocturnal awakenings were associated with suicide risk in patients with depression [48]. Other hypotheses, including the involvement of hyperactivity of the hypothalamic-pituitary-adrenal axis and hyperactivity of the noradrenergic system, have also been described as potential pathophysiological mechanisms of increased suicide risks in patients with depression with sleep disturbance [49]. Because both hypotheses appear to be related to responses to stressful events, it is difficult to establish a causal relationship, and more evidence is needed to test these hypotheses.

SDs, as one of the key symptoms of depression, may be the reason that people with depression seek an explanation in the first place and is one of the few proven risk factors for suicide [50]. MDD is closely associated with suicidal tendencies, with up to 3% of affected individuals ultimately dying by suicide [51]. In summary, our study showed that the suicide risk in individuals with SDs was 1.429 times higher than in those with no SDs. People with PDD exhibited 7.195 times the risk of suicide compared with those without PDD. SD patients with PDD exhibited 1.174 times the risk of suicide compared with those with no SDs. PDD in the presence of SDs was related to the highest risk of suicide.

This study has several limitations worth considering. First, similarly to previous studies using the NHI research database, we were unable to assess genetic factors, environmental factors, severity, or psychological assessments in patients with SDs or to classify SD severity and duration, illness duration, or the types of previous treatments used because these data were not recorded in the NHIRD [52]. Second, there was some unexplained heterogeneity in the data, and the lack of uniform diagnostic criteria may reduce comparability between studies. Third, missing data are present in nearly all studies, even in well-designed and controlled studies. Missing data can reduce the statistical power of a study and can produce biased estimates. Finally, we did not assess the presence of intellectual disability. Therefore, our results are a collection of potentially unstable estimates that are unlikely to be replicated. A more robust method is needed to validate our findings.

## 5. Conclusions

This study revealed that PDD affected SDs and was related to the highest risk of suicide, which warrants consideration. Patients suffering from SDs were at greater risk of suicide than patients with no SDs. People with PDD exhibited a greater risk of suicide than those without PDD. SD patients with PDD exhibited a greater risk of suicide than patients with no SDs. PDD in the presence of SDs had the greatest impact on suicide risk. Therefore, in addition to efforts to avoid suicidal events, healthcare providers should also be aware of PDD in the presence of SDs and the risk of suicide.

## Figures and Tables

**Figure 1 ijerph-19-13169-f001:**
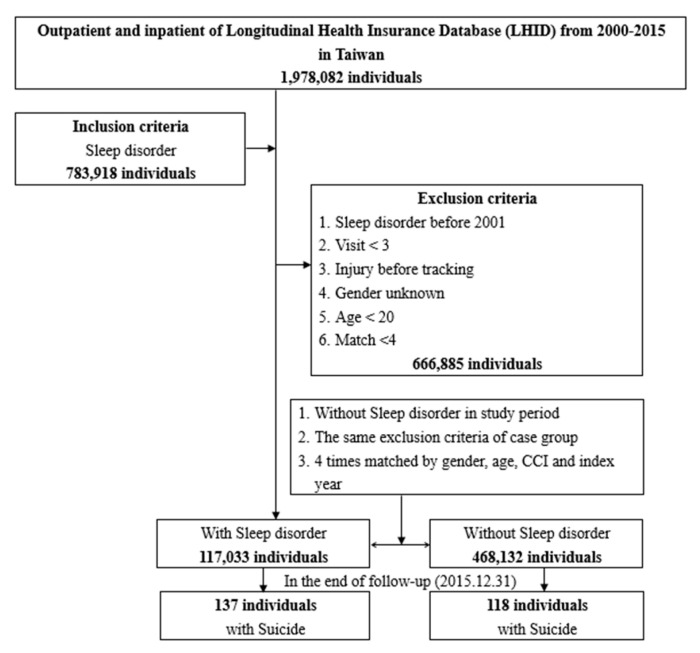
Flowchart of study sample selection from the National Health Insurance Research Database in Taiwan.

**Table 1 ijerph-19-13169-t001:** Characteristics of study participants.

Sleep Disorder	Total		With		Without		*p*-Value
Variables	*n*	%	*n*	%	*n*	%	
Total	585,165	100	117,033	20.0	468,132	80.0	
Gender							>0.999
Female	302,935	51.8	60,587	51.8	242,348	51.8	
Male	282,230	48.2	56,446	48.2	225,784	48.2	
Age (mean ± SD, year)	53.34 ± 15.74		53.43 ± 15.9		53.32 ± 15.69		0.089
CCI	1.64 ± 2.25		1.97 ± 2.41		1.56 ± 2.2		<0.0001
Age group (years)							0.592
20–39	128,943	22.0	25888	22.1	103055	22.0	
40–64	323,232	55.2	64309	54.9	258923	55.3	
≥65	132,990	22.7	26836	22.9	106154	22.7	
Low income							<0.0001
Without	580,429	99.2	115,740	98.9	464,689	99.3	
With	4736	0.8	1293	1.1	3443	0.7	
Diabetes mellitus							<0.0001
Without	455,183	77.8	87,578	74.8	367,605	78.5	
With	129,982	22.2	29,455	25.2	100,527	21.5	
Hypertension							<0.0001
Without	374,320	64.0	67,851	58.0	306,469	65.5	
With	210,845	36.0	49,182	42.0	161,663	34.5	
Persistent depressive disorder							<0.0001
Without	536,620	91.7	89,743	76.7	446,877	95.5	
With	48,545	8.3	27,290	23.3	21,255	4.5	
Chronic kidney disease							<0.0001
Without	555,977	95.0	110,516	94.4	445,461	95.2	
With	29,188	5.0	6517	5.6	22,671	4.8	
Heart failure							<0.0001
Without	546,859	93.5	108,065	92.3	438,794	93.7	
With	38,306	6.5	8968	7.7	29,338	6.3	
Chronic obstructive pulmonary disease and allied conditions							<0.0001
Without	377,009	64.4	67,670	57.8	309,339	66.1	
With	208,156	35.6	49,363	42.2	158,793	33.9	
Disorders of lipoid metabolism							<0.0001
Without	392,131	67.0	70,870	60.6	321,261	68.6	
With	193,034	33.0	46,163	39.4	146,871	31.4	
Diffuse diseases of connective tissue							<0.0001
Without	560,708	95.8	109,822	93.8	450,886	96.3	
With	24,457	4.2	7211	6.2	17,246	3.7	
Season							0.017
Spring (3–5)	14,820	2.5	2939	2.5	11,881	2.5	
Summer (6–8)	14,475	2.5	2732	2.3	11,743	2.5	
Autumn (9–11)	14,360	2.5	2832	2.4	11,528	2.5	
Winter (12–2)	541,510	92.5	108,530	92.7	432,980	92.5	
Location							0.004
Northern Taiwan	306,365	52.4	61,408	52.5	244,957	52.3	
Middle Taiwan	95,835	16.4	23,692	20.2	72,143	15.4	
Southern Taiwan	166,792	28.5	29,151	24.9	137,641	29.4	
Eastern Taiwan	12,756	2.2	2111	1.8	10,645	2.3	
Missing data	3417	0.6	671	0.6	2746	0.6	
Urbanization level							0.086
1 (Highest)	175,076	29.9	35,910	30.7	139,166	29.7	
2 (Second)	185,234	31.7	37,643	32.2	147,591	31.5	
3 (Third)	172,603	29.5	34,492	29.5	138,111	29.5	
4 (Lowest)	48,096	8.2	8161	7.0	39,935	8.5	
Missing data	4156	0.7	827	0.7	3329	0.7	
Level of care							<0.0001
Medical center	53,116	9.1	13,597	11.6	39,519	8.4	
Regional hospital	70,499	12.0	17,235	14.7	53,264	11.4	
Local hospital	68,124	11.6	15,135	12.9	52,989	11.3	
Clinic	393,426	67.2	71,066	60.7	322,360	68.9	

*p*-value (categorical variable: chi-squared/Fisher exact test; continuous variable: *t*-test).

**Table 2 ijerph-19-13169-t002:** Factors relared to suicide at the end of the follow-up, determined using Cox regression.

Variables	Adjusted HR	95% CI	*p*-Value
Sleep disorder			
Without	Reference		
With	1.429	1.073–1.905	0.015
Gender			
Female	Reference		
Male	1.297	1.001–1.681	0.049
Age (years)	0.946	0.935–0.957	<0.0001
CCI	0.906	0.839–0.979	0.012
Low income			
Without	Reference		
With	0.809	0.257–2.543	0.717
	Diabetes mellitus		
Without	Reference		
With	1.114	0.784–1.583	0.546
Hypertension			
Without	Reference		
With	1.123	0.809–1.558	0.489
Persistent depressive disorder			
Without	Reference		
With	7.195	5.378–9.626	<0.0001
Chronic kidney disease			
Without	Reference		
With	1.016	0.559–1.845	0.960
Heart failure			
Without	Reference		
With	1.536	0.949–2.486	0.080
Chronic obstructive pulmonary disease and allied conditions			
Without	Reference		
With	1.064	0.811–1.396	0.653
Disorders of lipoid metabolism			
Without	Reference		
With	0.735	0.538–1.003	0.053
Diffuse diseases of connective tissue			
Without	Reference		
With	0.793	0.460–1.367	0.403
Season			
Spring (3–5)	Reference		
Summer (6–8)	0.855	0.426–1.715	0.659
Autumn (9–11)	0.902	0.455–1.787	0.7668
Winter (12–2)	0.063	0.036–0.108	<0.0001
Location	Had multicollinearity with urbanization level		
Northern Taiwan	Had multicollinearity with urbanization level		
Middle Taiwan	Had multicollinearity with urbanization level		
Southern Taiwan	Had multicollinearity with urbanization level		
Eastern Taiwan	Had multicollinearity with urbanization level		
Urbanization level			
1 (Highest)	Reference		
2 (Second)	0.872	0.634–1.201	0.402
3 (Third)	0.944	0.685–1.303	0.728
4 (Lowest)	1.351	0.875–2.087	0.175
Hospital level			
Medical center	15.208	9.027–25.623	<0.0001
Regional hospital	21.651	13.306–35.229	<0.0001
Local hospital	23.282	14.230–38.091	<0.0001
Clinic	Reference		

*p* < 0.05 (categorical variable: chi-squared/Fisher exact test; continuous variable: *t*-test), HR = hazard ratio.

**Table 3 ijerph-19-13169-t003:** Factors related to suicide at the end of the follow-up, stratified by the variables listed in the table using Cox regression.

Sleep Disorder	Adjusted HR	95%CI	*p*-Value
Variables			
Total	1.429	1.073–1.905	0.015
Gender			
Female	1.172	0.814–1.686	0.394
Male	2.104	1.330–3.329	0.002
Age group (years)			
20–39	1.053	0.620–1.789	0.849
40–64	1.924	1.293–2.862	0.001
≥65	1.211	0.626–2.343	0.569
Low income			
Without	1.484	1.112–1.979	0.007
With	0.000		1.000
Diabetes mellitus			
Without	1.363	0.978–1.900	0.067
With	1.868	1.053–3.312	0.033
Hypertension			
Without	1.655	1.140–2.401	0.008
With	1.250	0.797–1.961	0.330
Persistent depressive disorder			
Without	1.174	0.820–1.680	0.382
With	2.050	1.350–3.112	0.001
Chronic kidney disease			
Without	1.483	1.104–1.993	0.009
With	1.299	0.379–4.451	0.677
Heart failure			
Without	1.447	1.068–1.959	0.017
With	1.686	0.717–3.967	0.231
Chronic obstructive pulmonary disease and allied conditions			
Without	1.769	1.207–2.592	0.004
With	1.215	0.793–1.861	0.371
Disorders of lipoid metabolism			
Without	1.715	1.207–2.437	0.003
With	1.145	0.702–1.867	0.587
Diffuse diseases of connective tissue			
Without	1.526	1.137–2.046	0.005
With	1.022	0.284–3.677	0.974
Season			
Spring (3–5)	1.061	0.374–3.007	0.912
Summer (6–8)	2.614	0.834–8.196	0.099
Autumn (9–11)	7.259	1.824–28.884	0.005
Winter (12–2)	1.244	0.903–1.713	0.183
Urbanization level			
1 (Highest)	1.502	0.900–2.505	0.119
2 (Second)	1.085	0.632–1.862	0.767
3 (Third)	2.071	1.219–3.518	0.007
4 (Lowest)	1.255	0.531–2.968	0.604
Hospital level			
Medical center	1.372	0.731–2.575	0.324
Regional hospital	1.463	0.927–2.308	0.102
Local hospital	1.371	0.820–2.290	0.229
Clinic	3.319	1.160–9.498	0.025

PYs = person-years; adjusted HR = adjusted hazard ratio, adjusted for the variables listed in the Cox sheets; CI = confidence interval; diabetes mellitus: ICD-9-CM 250; hypertension: ICD-9-CM 401–405; depression: ICD-9-CM 296.2–296.3, 296.82, 300.4, 311; insomnia: ICD-9-CM 780.52; stroke: ICD-9-CM 430–438; dementia: ICD-9-CM 290, 294.1, 331.0; chronic kidney disease: ICD-9-CM 585.

## Data Availability

Data are available from the National Health Insurance Research Database (NHIRD) published by the Taiwan National Health Insurance (NHI) Administration. Due to legal restrictions imposed by the government of Taiwan concerning the “Personal Information Protection Act,” data cannot be made publicly available. Requests for data can be sent as a formal proposal to the NHIRD (https://dep.mohw.gov.tw/DOS/lp-2506-113.html, accessed on 15 July 2022).

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
