# Peer review of "Persistent Depressive Disorder-Related Effect of Sleep Disorder on the Highest Risk of Suicide in Taiwan, 2000–2015"

_ijerph, 2022, doi:10.3390/ijerph192013169_

Round 1

Reviewer 1 Report

The descriptions are not consistent throughout the manuscript, and the manuscript cannot be published in the current status.

1.         (Abstract, line 25-26) The sentence is bizarre as an English sentence.

2.         (Abstract, line 27)Does “injured” mean suicide?

3.         (Abstract, line 28)Co co

4.         (Abstract, line 29)It is rate to write about statistical software in Abstract because they are not important information.

5.         (Results, Abstract)Cox regression analysis outputs results about hazard ratio, and it is different from risk ratio. Therefore, I think risk needs to be replaced with hazard.

6.         (Introduction)Could you add descriptions about “depressive neurosis effect of SD” in the Introduction? Sleep disorder and depressions are explained, but depressive neurosis effect of SD is merely mentioned in the Introduction. In addition, the authors should cite previous studies investigating an association between depressive neurosis effect of SD and suicide.

7.         (Methods) In the Abstract and Introduction, it is written that the aim is to investigate an effect on suicide. However, unintentional injuries are included in the outcome. If you include unintentional injuries in the outcome, Abstract and Introduction need to be revised accordingly.

8.         (Methods)You should write about censoring. The final observed time was the censoring day?

9.         (Table1)What are the definitions of urbanization level and low income?

10.     (Results, line 149) logistic regression? In addition, risk in the paragraph needs to be corrected.

11.     (Results, lines 163-165)Injuries are written in the descriptions, but suicide is written in the title of Table 4.

12.     (Results)It seems that you just investigated an association between SD and injury. It is not consistent with “an association between depressive neurosis effect of SD and suicide” written in the Introduction.

13.     (Discussion, lines 268-269)It is difficult to understand the sentence.

14.     (Discussion)Please write about implications of the study findings.

Reviewer 2 Report

This manuscript deals with the sleep disorder's possible association with suicide and what authors call depressive neurosis. As it is a rather atavistic term I believe authors should change it, primarily as the main body of the introduction and discussion deal with depression (a specific disorder diagnosed by psychiatrists). I know authors used ICD-9, but we are almost at ICD-11. I do not know if the registry noted MDD, PMDD, PDD, SAD, bipolar depression, something other, or plain depressivity (not a diagnosis).

Besides that, it is a reasonably straightforward study on a rather large cohort readily available for different retrospective analyses.

The main focus of the article is on the data presented in tables, which are large (Tables 1 and 2 could possibly be merged into one, although that would be a huge one).

The introduction is satisfactory, as is most of the methodology. However, I was unable to find the data on correction for regression analyses performed (especially multiple ones) and if the significance still holds after the correction.

The discussion is solid, with main points acknowledged as limitations.

In the end, the results are noteworthy and may guide further (more specified) research in the right direction, which is the main strength of this research.

Of smaller issues, references are not in the same style.

Reviewer 3 Report

A better understanding of the depressive neurosis effect of sleep disorders (SD) on the risk of suicide can aid, however, currentlyealth, efforts. Currently, there are limited longitudinal observational studies on the relationship between the depressive neurosis effect of SD on the risk of suicide. Therefore, in the present study, the Authors evaluated the depressive neuromost significantect of SD on the biggest risk of suicide and used the Ministry of Health and Welfare's National Health Insurance Research Database (NHIRD) to track whether the depressive neurosis effect of SD on the risk of suicide from 76 2000-2015 in Taiwan through long-term follow-up.

Overall, I found this study timely, original, well conducted, and scientifically sound. However, I have some suggestions aimed at improving the quality of the paper, and these are outlined below:

1) In the introduction, a brief note on the risk factors of suicidal ideation and behavior should be added with appropriate references (see dois 10.3390/brainsci10090591 and 10.30773/pi.2019.0171).

2) There isn't comprehensive information on inclusion and exclusion criteria. Please, add this part. For example, was the presence of an intellectual disability assessed, and how? The authors also wrote, "...we included people diagnosed with SD (ICD-9-CM). Codes: 780.5, 780.50". But, please, be more specific on which kind of sleep disorder was more prevalent, the severity, the duration, and so on.

3) I believe that in the Tables depicting clinical information on participants, other information should be added (with illness durations, kind of previous treatments, and so on).

4) Besides, I suggest adding in the table the complete statistics rather than a simple "p" value. Have also the Authors considered adding effect sizes?

5) I suggest improving the English language with the help of a native speaker.

Reviewer 4 Report

The manuscript addresses the relationship between sleep disorders, depression, and suicide in Taiwan.

It is not clear why the authors choose to use the term depressive neurosis instead of others like dysthymia, more in line with what is indicated in the ICD-9.

The introduction is relevant to the purpose of the manuscript.

In the study design in line 97, the term injured appears, which has not been clarified what it implies. It is not clear why to include accidents and self-harm in a study on suicide. It is a better option just to leave deaths by suicide.

It is not clear whether the result of the injury mechanisms was death or just injuries. It is necessary to disaggregate the data based on the fatal or non-fatal results of the injuries. The foregoing is based on what was declared by the authors in the title of the manuscript and the aim of the study. It is not clear the aims sought by including comorbidities in the statistical analysis.

The results on lines 128 and 127, present the age of the study participants in a similar way, either with arithmetic means or with percentages by age range.

In the tables of results, include in all of them the value of the statistical test used, in addition to the p-value.

In section 3.4 Comparison between the incidence and risk of injury of the sleep disorder and control cohorts, eliminate the statistical data and leave only the increase in risk. The statistical data can be read in table 4.

The discussion is the section of the manuscript with the greatest problems. The authors present many results associated with sleep disorders, depressive neuroses, injuries, and suicide. However, these results are rarely discussed. What is the point of including the variables of health, injuries, and medical attention if they will not be discussed? It is necessary to deepen the discussion of the data provided by the study.

In a few words, it is necessary for the authors to define the characteristics of the manuscript, whether they adhere to the proposed objective or generate new objectives that allow justifying the large number of analyzes they carry out.

The potential of the study is important due to the large number of patients, their follow-up, and the possible risk factors associated with suicide.

Round 2

Reviewer 1 Report

Thank you for the revision. I have a few comments.

1.      (Abstract, line 26)  It is written that "investigate whether PDD affects sleep disorder" as an objective, but its results are not shown in the Abstract. Did you investigate whether PDD affects sleep disorder or not ?

2.      (Abstract, line 37) The word "depressive neurosis" appeared suddenly.

3.      (Table 1)  Variables' names are positioned bizarrely. Please rearrange the table. For example, low income is positioned in the middle of the table.

4.      (Table 3)  If multicollinearity existed, how about removing it from the table ?

Reviewer 3 Report

Dear Authors, the paper is much improved and worthy of publication 

Reviewer 4 Report

The manuscript is much more transparent and understandable. It meets the proposed aim.

Some minor errors persist in the document.

In Table 2, in Disorders of lipoid metabolism, the p-value is 0.053, which seems to be a typo. Review it.

In Table 3, the letters in bold in the p-value column are absent in Chronic obstructive pulmonary disease and allied conditions, they are wrong in Season, Urbanization level, and Hospital level.

In the discussion, the term Depressive neurosis appears in lines 254, 314, 315, 316, and 317, change it to PDD.
